# Characterization of Viral miRNAs during Adenovirus 14 Infection and Their Differential Expression in the Emergent Strain Adenovirus 14p1

**DOI:** 10.3390/v14050898

**Published:** 2022-04-26

**Authors:** Eric R. McIndoo, Hailey M. Burgoyne, Hyung-Sup Shin, Jay R. Radke

**Affiliations:** 1Research Section, Boise VA Medical Center, Idaho Veterans Research and Education Foundation, Boise, ID 83642, USA; eric.mcindoo@va.gov (E.R.M.); hailey.burgoyne@va.gov (H.M.B.); 2College of Southern Nevada, Henderson, NV 89002, USA; ts.tony4sure@gmail.com; 3Department of Biological Sciences, Boise State University, Boise, ID 83725, USA

**Keywords:** human adenovirus, Adenovirus 14, Adenovirus 14p1, VA RNA, mivaRNA

## Abstract

Human adenoviruses (HAdV) express either one or two virus-associated RNAs (VA RNAI or VA RNAII). The structure of VA RNA resembles human precursor microRNAs (pre-miRNA), and, like human pre-miRNA, VA RNA can be processed by DICER into small RNAs that resemble human miRNA. VA RNA-derived miRNA (mivaRNA) can mimic human miRNA post-transcriptional gene repression by binding to complementary sequences in the 3′ UTR of host mRNA. HAdV14 is a member of the B2 subspecies of species B adenovirus, and the emergent strain HAdV14p1 is associated with severe respiratory illness that can lead to acute respiratory distress syndrome. Utilizing small RNA sequencing, we identified four main mivaRNAs generated from the HAdV14/p1 VA RNA gene, two from each of the 5′ and 3′ regions of the terminal stem. There were temporal expression changes in the abundance of 5′ and 3′ mivaRNAs, with 3′ mivaRNAs more highly expressed early in infection and 5′ mivaRNAs more highly expressed later in infection. In addition, there are differences in expression between the emergent and reference strains, with HAdV14 expressing more mivaRNAs early during infection and HAdV14p1 having higher expression later during infection. HAdV14/p1 mivaRNAs were also shown to repress gene expression in a luciferase gene reporter system. Our results raise the question as to whether differential expression of mivaRNAs during HAdV14p1 infection could play a role in the increased pathogenesis associated with the emergent strain.

## 1. Introduction

Human adenoviruses (HAdVs) are non-enveloped viruses with a linear double-stranded DNA genome of ~36,000 bp. HAdVs are grouped into seven species (A–G) and consist of over 100 different types (consecutively numbered) based on genomic sequencing. HAdVs can infect the respiratory, gastrointestinal, and urinary tracts as well as the conjunctiva. Overall, HAdV usually results in mild, self-limited infections in immunocompetent individuals. HAdV14 is a member of the B2 subgroup of adenovirus and, as such, predominately causes kidney and urinary tract infections but can also be associated with respiratory infections. Outbreaks of emergent strains of Ad have resulted in severe and sometimes fatal infections in otherwise healthy people. In the past fifteen years, an emergent strain of HAdV14, HAdV14p1, emerged first in the U.S. and subsequently throughout the world [1,2,3,4,5,6]. HAdV14p1 was first identified as the causative agent of an outbreak of acute respiratory distress syndrome (ARDS) in U.S. military populations [7]. HAdV14p1 is 99.7% identical to HAdV14 yet displays increased lung pathogenesis in the Syrian hamster that models HAdV14p1 pathogenesis in humans [8,9]. Why HAdV14p1 has increased pathogenesis is still unclear.

The HAdV genome encodes roughly 40 proteins. In addition to protein encoding RNAs, non-coding RNAs (ncRNA) are also present in HAdV-infected cells [10,11,12]. Viral-associated RNA (VA RNA) are encoded by the HAdV genome and are the most abundant viral ncRNA found in infected cells [13,14]. All HAdV encode at least one VA RNA transcript ~160 nt in length, and approximately 80% of HAdV serotypes encode at least 2 VA RNAs (VA RNA-I and VA RNA-II), with VA RNA-I encoded by all HAdV serotypes [15,16]. VA RNAs are required for efficient viral replication, as the deletion of VA RNA-I reduces viral titer by 20-fold and the deletion of both VA RNA-I and VA RNA-II results in a 60-fold decrease in viral titer [17,18]. Unlike protein encoding HAdV RNAs, VA RNAs are transcribed by cellular RNA polymerase III and are expressed throughout the viral replication cycle [13]. 

VA RNA forms a complex secondary structure that is critical for its biological function [19,20,21]. Overall, despite their low sequence conservation among HAdV, VA RNAs share similar secondary structure and organization of the apical stem, central domain, and terminal stem [16,21]. The crystal structure of HAdV-2 VA RNA-I reveals a sharply bent, coaxially stacked, wobble-enriched, and pseudoknot-anchored molecule to allow VA RNA to interfere with nearly all host systems that interact with double-stranded RNA (dsRNA) [19]. The best characterized activity of VA RNA is its inhibition of protein kinase R (PKR) through the ability of VA RNA-I to bind and inhibit PKR dimerization [22,23,24]. PKR is part of the cellular innate immune response to viral infection, and, by sensing dsRNA produced during viral infection, PKR activation results in the global inhibition of cap-dependent protein synthesis. VA RNA also binds to 2′-5′ Oligoadenylate Synthetase 1 (OAS1), which results in the activation of OAS1 [25]. While the interaction of full-length VA RNA activates OAS1, the binding of cleaved VA RNAI (produced following DICER cleavage) to OAS1 inhibits OAS1 activity [11,26,27]. Interestingly, the cleaved VA RNAI has a higher affinity for OAS1 than full-length VA RNAI, suggesting that the inhibitory activity of cleaved VA RNAI may be the predominant activity of VA RNAI on OAS1 [28]. VA RNAI binds to the retinoic acid-inducible gene I (RIG-I). The triphosphorylyated 5′-end nucleotide of VA RNA triggers RIG-I signaling, which results in an increased type I interferon response [29,30,31,32]. These roles in modulating the cellular innate immune response to HAdV have been long understood.

Recently, new roles for VA RNAI in regulating cellular miRNA processes have been elucidated. Following transcription, VA RNAI is exported from the nucleus to the cytoplasm by Exportin 5 (Exp5). Exp5 exports RNA molecules (e.g., pre-miRNA and tRNAs) that contain a ‘mini-helix’ from the nucleus [33,34,35,36]. VA RNAI contains a ‘mini-helix’ at the end of the terminal stem, which results in VA RNA being a direct competitor with cellular small RNAs for interaction with Exp5 [37]. In addition to competing with pre-miRNA for nuclear export through Exp5, once in the cytoplasm, VA RNAs are cleaved by DICER, like pre-miRNA, and as VA RNA expression increases, more VA RNAs are associated with DICER than host pre-miRNA, resulting in decreased host miRNA expression [27,38,39]. DICER processes VA RNA into small RNAs that resemble host miRNA (called mivaRNA) that can be incorporated into the RNA-induced silencing complex (RISC) [37,38,40,41,42]. RISC association allows for mivaRNA to bind and repress target mRNA expression [38,40,41,42,43].

Group B2 HAdV members encode only the VA RNAI gene. Bioinformatic analysis of the HAdV14 genome showed that the VA RNAI gene of Ad14 is ~98% identical to the other group B2 members HAdV11, HAdV34, and HAdV35 [44]. In this report, we further characterize the VA RNA gene, identifying the intragenic promoter elements, transcriptional stop sites, potential OAS1 consensus sites, and GGGU/ACCC sites. Small RNA-seq was used to identify the mivaRNA produced during infection. Despite 100% sequence identity of the VA RNAs, differential expression of the mivaRNA exists between HAdV14- and HAdV14p1-infected A549 cells. We discuss the possibility that this differential mivaRNA expression could play a role in the pathogenesis of HAdV14p1.

## 2. Materials and Methods

### 2.1. Cells and Viruses

A549 cells (CCL-185, ATCC, Manassas, VA, USA) were grown at 37 °C and 5% CO_2_ in DMEM supplemented with 5% heat-inactivated bovine calf serum, 100 U/mL penicillin, 100 µg/mL streptomycin, and 2 mM L-glutamine. A549 cells were monitored for mycoplasma contamination by PCR and validated by short tandem repeat markers (STR Authentication) (ATCC). HAdV14 deWit was obtained from ATCC, and HAdV14p1 (isolate 1986T) was obtained from the United States Navy [34,35,36]. Viruses were propagated and plaque titered (plaque-forming units (pfu)/mL) in A549 cells.

### 2.2. Infection of A549 Cells and Isolation of Total Cellular RNA

A549 cells were infected with either HAdV14 or HAdV14p1 at a multiplicity of infection (MOI) of 10 pfu/cell in suspension for 1 h at 37 °C, after which cells were plated and allowed to adhere until collected. Adherent and non-adherent cells were collected at 6, 12, 24, 36, and 48 h post-infection. Total RNA was isolated using miRNeasy kit (Qiagen, Germantown, MD, USA) with on column DNase treatment. The total RNA in each sample was quantified using the Qubit 2.0 Fluorometer (Invitrogen, Carlsbad, CA, USA), and quality was measured using the RNA6000 Nano chip on the Agilent 2100 Bioanalzyer (Agilent Technologies, Santa Clara, CA, USA). All samples had an RNA integrity number greater than 7.

### 2.3. Small RNA Library Preparation, Sequencing and Data Analysis

TruSeq Small RNA library prep kit (Illumina, San Diego, CA, USA) was used to create sequencing libraries. Specifically, adapters were ligated using the 5′ phosphate and 3′ hydroxyl groups common to most mature miRNAs. After adapter ligation, samples were reverse transcribed and amplified. Finally, the libraries were size selected using a 6% polyacrylamide gel and concentrated using ethanol precipitation. Purified libraries were normalized and pooled to create a double-stranded cDNA library and were sequenced on the Illumina MiSeq to render 50 base pair single end reads at the Loyola Stritch School of Medicine Genomics Facility. Adapter sequences were removed, and low-quality reads were trimmed from raw sequencing reads using Cutadapt (v. 1.11). The resulting reads were mapped to the HAdV14 deWit genome (GenBank accession number AY803294) in CLC Genomics Workbench (Qiagen).

### 2.4. Differential Expression Analysis

Differential expression was conducted with CLC Genomics Workbench. The VA RNA sequence was added to miRbase 22 and annotated to contain either total 5′ and 3′ reads or mivaRNA 5′A, 5′G, 3′A and 3′C seeds. Quantify miRNA 1.2 (CLC) was used with the following settings: allow for length-based isomiRs, no additional upstream bases, 5 additional downstream bases, no missing or mismatch bases, minimum sequence length = 18, maximum sequence length 25. Differential Expression for RNA-Seq 2.6 (CLC) was used for differential expression analysis with normalization set to Trimmed Mean of Means (TMM).

### 2.5. HAdV-14 mivaRNA 3′ UTR Luciferase Reporter Assays

The complement of the either the 5′ or 3′ seed sequences of Ad14 VA RNA were cloned into the multiple cloning site of pmirGLO Dual Luciferase Vector (Promega, Madison, WI, USA). Constructs were sequenced to confirm the correct orientation of the complimentary HAdV14 VA RNA seed sequences. Plasmids were purified using PureLink HiPure Plasmid Maxi Prep (Invitrogen) and transfected into A549 cells using Lipofectamine LTX (Invitrogen). Twenty-four hours after transfection, cells were infected with Ad14 at an MOI of 10. After an additional 24 h, luciferase activity was determined using a dual-luciferase assay (Promega). Firefly luciferase activity was normalized to Renilla luciferase activity. One-way ANOVA was followed by the Holm-Sidák test with *p* < 0.05 considered significant (Prism 9; GraphPad Software, San Diego, CA, USA).

### 2.6. Prediction of mivaRNA Host Gene Targets and Functional Enrichment Analysis

TargetScan and miRDB were used to predict cellular targets for mivaRNAs by seed sequences and filtered to remove duplicate targets. KEGG (https://www.genome.jp/kegg/, accessed on 2 May 2021) was used to predict the cellular pathways and processes potentially regulated by mivaRNA–target mRNA interaction.

### 2.7. RNA-Seq Library Preparation, Sequencing, Differential Expression Analysis, and Bioinformatic Analysis

Illumina stranded mRNA libraries were made from the total RNA preps from A549-infected cells. Sequencing was performed at University of Oregon Genomics & Cell Characterization Core (Eugene, OR, USA) on an Illumina HiSeq4000 set for pair ends 100 bp reads and demultiplexed. Reads were trimmed with Trim Reads 2.4 (CLC) before RNA-seq Analysis 2.21 (CLC) was performed, mapping to the human genome (Hg38) with a mismatch cost of 2, insertion cost of 3, deletion cost of 3, and length/similarity fractions set to 0.8. Paired reads were counted as a single read and expression value was set to reads per kilobase million (RPKM). Differential expression for RNA-Seq 2.4 (CLC) using gene expression values from RNA-seq analysis was performed and the results were filtered to the predicted mivaRNA targeted genes. Ingenuity Pathway Analysis (IPA) (Qiagen) was used for functional enrichment with the differential RNA-seq analysis from HAdV14-infected cells at 36 hpi vs. uninfected cells. FDR was set to 0.05 with absolute expression fold changes being >1.5 and expression intensity being >1.

## 3. Results

### 3.1. Analysis of HAdV14/HAdV14p1 VA RNA Gene Sequence and Secondary Structure

Like other HAdV group B2 members, HAdV14 and HAdV14p1 genomes encode only 1 VA RNA sequence that resembles VA RNA-I at nucleotides 10,452–10,613 [44]. Houng and colleagues have shown that the sequences of the VA RNA genes of HAdV14 and HAdV14p1 are 100% identical [9]. Analysis of the VA RNA-I sequence reveals two intragenic promoter elements (Box A (+13–23) and Box B (+55–65)) that are required for RNA polymerase III transcription (Figure 1A). VA RNA-I transcription in many serotypes has been shown to be initiated at one of two potential transcription start sites, either a G(+1) or three nucleotides upstream at A(−3) [16,27]. The HAdV14 VA RNA-I gene contains both possible start sites. Termination sequences (T1A and T1B) are found at +152–161. Additionally, there is a backup termination sequence that is found at +192–198. The VA RNA gene contains the conversed GGGU (+33–36) and ACCC (+120–123) that form the central domain (Figure 1A,B). Figure 1B shows a predicted secondary structure for the HAdV14 VA RNA molecule consisting of an apical stem, central domain, and terminal stem. HAdV VA RNA I contains two OAS1 consensus sites (WW(N_9_)WG) found in the central domain [45,46]. HAdV14 contains two OAS1 consensus sites at +36–48 and +93–105; both are found in the central domain.

### 3.2. Identification and Temporal Expression of HAdV14 Small RNAs 

Previous studies have shown that the majority of small RNAs produced during HAdV infection map to the VA RNA gene(s). To examine the viral small RNAs produced during HAdV14 or HAdV14p1 infection, A549 cells were infected with either HAdV14 or HAdV14p1 (Figure 2). Four independent small RNA libraries from infection with either HAdV14 or HAdV14p1 per time point were sequenced. At 6 hpi, less than 1% of the reads mapped to the HAdV14 genome, then increased to ~20% at 24 hpi, then remained at ~15% at 36 and 48 hpi (Table 1). At all time points, >92% of all small RNAs that mapped to the HAdV14 genome mapped to VA RNA gene (Table 1), showing that the VA RNA gene is responsible for the vast majority of all HAdV14-encoded small RNAs. The diversity of the reads that aligned to the VA RNA gene increased during infection mainly by extensions or deletions at either the 5′ or 3′ ends of the small VA RNAs, with the total number of unique small VA RNAs similar between HAdV14 and HAdV14p1 (Appendix A). The majority of the reads that aligned to the VA RNA gene mapped to the 5′ and 3′ ends of the terminal stem, with each small VA RNA comprising more than 1% of the total small RNA pool sequenced (Table 2). The number of unique small RNAs that mapped to the VA RNA gene increased from 6 to 24 hpi and then stabilized through 48 hpi (Figure 3A). Examination of the read counts showed temporal expression of small RNAs produced from the 5′ or 3′ ends of the VA RNA gene. At 6 and 12 hpi, the 3′ small RNAs are predominant, while the 5′ small RNAs show increased expression 36 and 48 hpi (Figure 3B–F). The same temporal expression patterns are seen in HAdV14p1-infected cells (Figure 3B–F). The differences in read counts between HAdV14- and HAdV14p1-infected cells suggest that there may be differential expression of 5′ and 3′ small VA RNAs produced during infection. Differential expression analysis showed that, for the most part, 5′ small VA RNAs during HAdV14p1 infection were not differentially expressed compared to HAdV14 infection, except at 12 hpi. In contrast, 3′ small VA RNAs were differentially expressed during HAdV14p1 infection compared to HAdV14 infection, except at 24 hpi.

### 3.3. Identification of Potential HAdV14/14p1 mivaRNAs

While there is diversity in the small RNAs that map to the VA RNA gene, repression of target gene expression involves the binding of miRNA to the target mRNA through the miRNA seed sequence, nucleotides 2–8 of the miRNA, and its complement on the target gene mRNA. Therefore, the seed sequences provide a better metric for both identifying and quantitating HAdV mivaRNAs. Unique miRNA seed sequences were identified for each small VA RNA molecule (Table 3). The total number of unique seed sequences increased during HAdV14 or HAdV14p1 infection and plateaued at 24 hpi (Figure 4A). The number of unique seed sequences was similar in both HAdV14- and HAdV14p1-infected cells (Figure 4A). To identify potential HAdV14/14p1 mivaRNAs, we identified the seed sequences that had total read counts >1000 and were found in small RNAs 20–33 nucleotides long. Eleven seed sequences were identified that met those criteria, and all of them mapped to either the 5′ or 3′ terminal ends of the VA RNA (Figure 4B). Based on the read counts, there are four main mivaRNAs produced during HAdV14/14p1 infection, two from both the 5′ and 3′ ends. The mivaRNAs from the 5′ end start at either the A (−3) or G (+1) and have been named mivaRNA 5′A or mivaRNA 5′G. At the 3′ end, the predominant mivaRNAs start at either C (138) or A (139) and have been designated mivaRNA 3′C and 3′A, respectively (Figure 4B). All four of the predominant mivaRNAs average 20 to 24 nucleotides in length, with the 5′ mivaRNAs showing the greatest length diversity, out to 50 nucleotides long (Appendix A). Overall, >95% of the reads are between 20 and 24 nt. Using our read counts (Table 2) and unique seed sequence data (Table 3), we predict that there are five dominant DICER cleavage sites in HAdV-14/14p1 VA RNA. These predicted cleavage sites generate greater than 98% of the unique seed sequences that map to the VA RNA gene (Figure 4C). We predict at least four secondary cleavage sites that generate the unique seed sequences in Table 3 that have more than 1000 total counts (Figure 4C). 

### 3.4. Differential Expression of mivaRNA during HAdV14 and HAdV14p1 Infection 

Infection with HAdV14 or HAdV14p1 resulted in differential expression of 3′ small VA RNAs (Figure 3). With the dominant mivaRNAs identified, we performed differential expression analysis of each mivaRNA over time in HAdV14p1-infected vs. HAdV14-infected cells. There was no differential expression of mivaRNA 5′A during infection (Figure 5A). However, mivaRNA 5′G (Figure 5B) showed differential expression that was lower at 6 and 12 hpi in HAdV14p1-infected cells and then was higher in HAdV14p1-infected cells at 48 hpi. In HAdV14p1-infected cells, both of the 3′ mivaRNAs (3′C and 3′A) showed a lower expression at 6 and 12 hpi compared to HAdV-14-infected cells (Figure 5C,D). At later times (36 and 48 hpi), mivaRNA 3′C and 3′A were both expressed at higher levels during HAdV14p1 infection than during HAdV14 infection. Overall, the data show that there are temporal mivaRNA expression differences between HAdV14- and HAdV14p1-infected cells.

### 3.5. Expression of HAdV14 mivaRNA and A549 miRNA Expression during Infection 

DICER processes pre-miRNAs and VA RNA into mature miRNA and mivaRNA, respectively [27,39]. During HAdV infection, it has been shown that VA RNAs can suppress miRNA biogenesis by competing with host pre-miRNAs for cleavage by DICER [38]. Infection with HAdV14p1 results in nearly a 50% reduction in cellular miRNA at 24 hpi that persists through full CPE (Table 4). To explore the relative abundance of HAdV14 mivaRNAs in relation to A549 miRNAs, the identified HAdV-14 mivaRNAs were added to the miRBase v22 list of human miRNAs to allow for counting of mivRNAs and host miRNAs in the CLC Genomics Workbench. At 6 hpi, only mivaRNA 3′C was one of the top 20 total miRNAs expressed (Table 5). Beginning at 12 hpi, mivaRNA 3′C, 5′A, and 5′G were found in the top six total miRNAs expressed and remained in the top six through full CPE at 48 hpi. After 6 hpi, all four predominant mivaRNAs were expressed at similar levels to the top cellular miRNAs. In general, increased expression of mivaRNAs resulted in decreased expression of all of the top 20 cellular miRNAs.

### 3.6. HAdV14 mivaRNA Are Functional miRNAs 

Since HAdV VA RNAs are processed by DICER into mivaRNAs, we hypothesized that the mivaRNAs can repress gene expression like cellular miRNAs. To test the ability of HAdV14 mivaRNA to function as miRNA, the complement of each of the four main mivaRNA seed sequences were cloned into the 3′ UTR of luciferase in the pmirGLO Dual Luciferase Vector. Each reporter vector was transfected into A549 cells 24 h prior to infection with HAdV-14 at an MOI of 10, and luciferase activity was determined 24 hpi. As shown in Figure 6, HAdV-14 infection resulted in the repression of luciferase activity from all four mivaRNAs at 24 hpi. The strongest repression was seen from the 3′ mivaRNAs that are expressed at higher levels at 6 and 12 hpi (Figure 3B). 

### 3.7. Prediction of mivaRNA Host Targets and Functional Enrichment Analysis

Having established that mivaRNAs are expressed at levels similar to cellular miRNAs (Table 5) and are capable of repressing gene expression (Figure 6) of target genes that have the complementary seed sequence in the 3′ UTR of their mRNA, we sought to determine if there was any potential for HAdV14 mivaRNAs to repress cellular genes. TargetScan and mirDB were used to predict potential host target mRNAs using the seed sequences identified (Figure 4B). All but one of the mivaRNAs are unique seed sequences that are not found in human miRNAs. The 3′-mivaRNA-CAAAAATCCAGGATACGGAATCGAGTCGTT encodes the same seed sequence as human miR-584c-3p and was excluded from further analysis. The remaining 10 mivaRNA seeds were used to predict potential gene targets. Overall, there were 2184 predicted genes that could be targeted by the Ad14 mivaRNAs (Appendix A). Of the 2184, 1919 were recognized by KEGG, and KEGG pathway analysis was performed on the predicted gene targets. The top pathways included many signal transduction pathways and cellular processes involved in viral infection, including metabolism, cytoskeleton/tight junctions/focal adhesion/cell adhesion molecules, mRNA transport/splicing/regulation, and cell death processes (Table 6 and Appendix A).

### 3.8. Analysis of Predicted Target Gene Expression by RNA-Seq

Since our in vitro studies showed that the mivaRNAs can functionally repress luciferase gene expression (Figure 6), we sought to determine if mivaRNA might regulate expression of predicted target genes. RNA-seq analysis was performed on the total RNAs from cells at all time points. Differential expression analysis was performed against uninfected control RNA libraries. The genes were filtered to the 1919 genes recognized by KEGG analysis, and genes that were down regulated by at least 1.5-fold with an FDR of <0.05 were considered significantly down regulated. As seen in Figure 7A, from 6 hpi to 24 hpi, down regulated target genes increased from ~200 to ~370 for both HAdV14- and HAdV14p1-infected cells. Between 24 and 36 hpi, the number of down regulated genes plateaued (HAdV14) or slightly decreased (HAdV14p1) from the 24 hpi number. At 48 hpi (full CPE), the number of down regulated genes doubled for both HAdV14 and HAdV14p1 from the 36 hpi time point. At both 36 and 48 hpi, there were more down regulated target genes in HAdV14-infected cells than in HAdV14p1-infected cells. From 6 hpi to 24 hpi, ≥78% of down regulated targets were shared between HAdV14- and HAdV14p1-infected cells (Figure 7B). This similarity to target repression decreased at 36 and 48 hpi. To gain a better understanding of how the down regulated gene expression might alter cellular pathways, we performed IPA with the 36 hpi differential gene expression data sets. This time point was selected, as there should be sufficient repression observed at 36 hpi, and the time course of viral infection has not resulted in 4+ CPE. As seen in Table 7, the top ten pathways enriched from our RNA-seq data are very similar between HAdV14- and HAdV14p1-infected cells. The two pathways that are not the same in the top ten are not far out of the top ten (Appendix A). Combined with the ability of mivaRNAs to functionally repress luciferase gene expression, these data suggest that mivaRNAs can regulate host gene expression during infection. However, further studies are needed to determine whether other viral genes or indirect effects of viral replication affect host gene expression.

## 4. Discussion

The adenoviral VA RNA gene is a multifunctional non-coding RNA molecule. VA RNAs are processed by DICER into mivaRNA that resemble and function like cellular miRNAs. While all human adenoviruses encode at least one VA RNA gene, the amount of sequence similarity between VA RNA genes from all human adenoviruses is limited between all human Ad, but is somewhat conserved within Ad groups. In this study, we sought to identify the mivaRNA produced during prototype HAdV14 deWit infection, assess predicted target gene expression, and compare mivaRNA expression with that of the emergent pathogenic strain HAdV14p1, using A549 cells that are permissive for HAdV14 and HAdV14p1 infection.

HAdV14, like other group B2 HAdV, only encode one VA RNA gene [44]. In HAdVs that only encode one VA RNA, that VA RNA resembles group C HAdV VA RNAI. Here, further bioinformatic analysis of the HAdV14 VA RNA gene has shown that the VA RNA gene contains both RNA Pol III Box A and B promoter sites as well as two internal terminator sequences and a backup terminator sequence. The HAdV14 VA RNA gene also encodes two potential OAS1 consensus activation sequences. Based on the predicted secondary structure of the HAdV14 VA RNA molecule, it includes a terminal stem, central domain, and apical stem. Overall, the bioinformatic results suggest that, like other HAdVs that only encode one single VA RNA gene, HAdV14 VA RNA resembles that of VA RNAI, rather than VA RNAII. 

To characterize the mivaRNA produced from the HAdV14 VA RNA gene, we infected A549 cells and sampled mivaRNA expression through all phases of the Ad14 infectious cycle from very early (6 hpi) all the way through to full CPE (48 hpi). At all times, >90% of all small RNA reads that mapped to the HAdV14 genome mapped specifically to the VA RNA gene. These results are consistent with those seen during Ad5 infection of IMR-90 cells [48]. Zhao and colleagues first observed that there was temporal expression of mivaRNA molecules during the infection cycle [48]. We also observed temporal expression of mivaRNAs during HAdV14 infection. Our data showed that early during infection, the 3′ mivaRNAs are the most abundant HAdV14 mivaRNAs, while between 24 and 36 hpi, the 5′ mivaRNAs begin to be more abundant. This is opposite of what was observed following HAdV2-infected IMR-90 cells. The reason for this difference is unclear. The difference could be virus-specific (HAdV2 vs. HAdV14), cell type-specific (lung fibroblast, IMR-90, vs. lung epithelial, A549), or a combination. Supporting this theory are studies by Kamel et al., which showed that infection of 293 cells with HAdV5, HAdV11 (a group B2 HAdV), and HAdV37 resulted in more 3′ mivaRNA expression early during infection [27]. The same temporal expression patterns were seen during HAdV14p1 infection, as was observed by Zhao [48]. When we normalized mivaRNA expression, we observed that there was a significant difference in the expression of total mivaRNAs between HAdV14 and HAdV14p1 over time. Specifically, early in the infection cycle, HAdV14 mivaRNAs showed higher expression than HAdV14p1. After 24 hpi, this pattern is flipped, with HAdV14p1 mivaRNAs expressed at a higher level than HAdV14. Previous studies in our lab and others have shown that there is no difference in the replication rates of HAdV14 or HAdV14p1 in either tissue culture cells or in vivo in the lungs of Syrian hamsters, thus eliminating differences in viral replication as the reason for the differential expression of mivaRNAs [8,49]. The studies here were not designed to test whether this mivaRNA expression pattern has any role in the increased pathogenesis seen during HAdV14p1 infection. Studies are underway to test this possibility using a Syrian hamster model of differences in HAdV14p1 vs. HAdV14 lung pathogenesis [8,50]. 

Our results identified the core seed sequences produced during Ad14/Ad14p1 infection as coming from both the 5′ and 3′ terminal stems. Minor variants are also produced. How these arise is unclear. One possibility is that RNA pol III synthesis of VA RNAs might have some minor wobble in its exact initiation site, as these minor mivaRNA species show very slight variations in either the 5′ or 3′ ends. It is possible that this could have an effect on expression, as small differences in the 5′ start can result in a completely different mivaRNA seed sequence. We identified a group of long mivaRNAs produced from the 3′ terminal stem that start between 8 and 4 nucleotides upstream of the predominant 3′ mivaRNA. Interestingly, the longest mivaRNA produced from the 3′ terminal stem has the exact same seed sequencing as hsa-miR-584c-3p. It has been well documented that HAdV mivaRNAs can regulate gene expression, like gene-encoded miRNAs [26,40,42]. Using a luciferase reporter vector, we have shown that both the 5′ and 3′ HAdV-14 mivaRNAs can suppress gene expression during viral infection. Our results confirm that the 3′ mivaRNAs are more abundantly expressed than 5′ mivaRNAs early during infection, as the 3′ mivaRNAs showed stronger repression of luciferase activity than the 5′ mivaRNAs. mivaRNAs have been shown to associate with RISC complexes, but not all mivaRNAs associate with RISC complexes at the same rate [27,38,40,42]. This could account for the difference in repressive activity seen between the mivaRNAs in our reporter assay. The association of HAdV14 mivaRNAs with the RISC complex was not a focus of these studies, but is under investigation. 

The ability of mivaRNAs to target and regulate host gene expression is an attractive theory for potential roles in HAdV replication. As a result, numerous studies have been conducted to determine potential target genes for mivaRNAs. Bioinformatic, microarray studies, and RNA-seq analysis have shown that many of the potential cellular gene targets for mivaRNAs are involved in DNA repair, cell growth, cell death, RNA metabolism, cell signaling pathways, and metabolism [26,42,48]. Our bioinformatic analysis (Table 6) on the predicted HAdV14 mivaRNA targets show similar cellular pathways to be potential targets, despite the lack of sequencing identity between HAdV14 seed sequences and HAdV2/5 seed sequences. Our IPA analysis (Table 7) also shows functional enrichment of the same cellular pathways from our RNA-seq data on HAdV14 mivaRNA target genes. This suggests that there is some conserved biological reason for mivaRNA expression during viral infection. However, studies have raised questions as to the role of mivaRNAs in viral replication. The mutation of the seed sequences of HAdV5 mivaRNAs had no impact on viral replication in HEK293 cells [51]. One underlying problem with studies (including these studies) on HAdV mivaRNAs is that they have been conducted in tissue culture cell lines rather than relevant primary cells or in a permissive animal model, which might clarify the role of mivaRNAs during infection in vivo. HAdV mivaRNAs have also been found in persistently infected lymphoid cells, suggesting a potential role in regulating lytic and persistent infection [52,53]. In addition to mivaRNAs from the VA RNA gene, another viral small RNA that maps to the Major Late Promoter region has been identified that regulates viral gene expression; we are in the process of exploring if there are other HAdV14 small viral RNAs [54].

In this study, we characterized the HAdV14 VA RNA gene and the mivaRNAs produced from it. We identified the predominant seed sequences and the cellular genes that the mivaRNAs can target. Our bioinformatic analysis using RNA-seq data of the expression of mivaRNA target genes showed that HAdV14 mivaRNA can regulate similar pathways seen from HAdV2/5 studies despite the lack of identity of the mivaRNA seed sequences. Despite 100% sequence identity later in infection, HAdV14p1-infected cells express significantly more mivaRNA than HAdV14-infected cells. Whether this difference has an impact on viral replication or pathogenesis is unknown. Due to a lack of studies in immunocompetent animal models, the role of HAdV mivaRNA in viral pathogenesis is largely unexplored. To answer questions about viral factors, such as mivaRNAs, we have developed an immunocompetent Syrian hamster model of HAdV14p1 lung pathogenesis that displays pathology similar to what is seen in the lungs of severely infected human patients [8,50]. We propose that the Syrian hamster model of HAdV14p1 lung pathogenesis could provide the model system to explore the roles of mivaRNA in viral replication and pathogenesis in vivo.

## Figures and Tables

**Figure 1 viruses-14-00898-f001:**
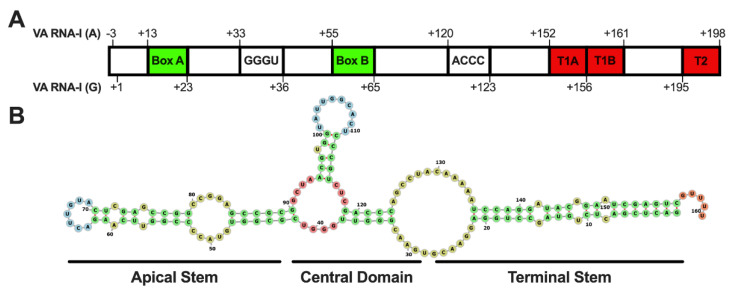
Human adenovirus 14 VA RNA gene and transcript secondary structure. (**A**) The HAdV14 VA RNA gene contains two predicted start sites A (−3) and G (+1). RNA Pol III Box A and B promoter regions (green), two internal terminator sequences (red, T1A and T1B), and a backup terminator sequence (T2) are indicated. Two complementary sequences (GGGU and ACCC) are also present. (**B**) A predicted secondary structure shows the regions that form the terminal stem, central domain, and apical stems (RNAStructure [47]).

**Figure 2 viruses-14-00898-f002:**
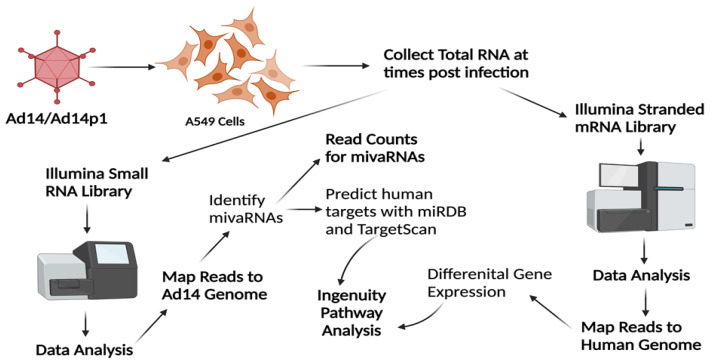
RNA-seq and bioinformatics workflow. A549 cells were infected with HAdV14 or HAdV14p1 at an MOI of 10:1 and total RNA was collected at various times post-infection. To identify HAdV14 mivaRNAs, Illumina small RNA-seq libraries were constructed, sequenced on MiSeq, and reads were mapped to the HAdV14 genome to identify HAdV14 mivaRNAs. Quantitation of reads was performed with CLC Genomics Workbench. TargetScan and miRDB were used to predict human mRNA targets based on seed sequences of the identified mivaRNAs. To examine gene expression in A549 cells during infection, Illumina stranded mRNA-seq libraries were constructed, sequenced on a NextSeq 2000, reads mapped to the human genome, and differential gene expression determined. Ingenuity Pathway Analysis was used to explore differential expression of mivaRNA predicted target genes. Created with BioRender.com.

**Figure 3 viruses-14-00898-f003:**
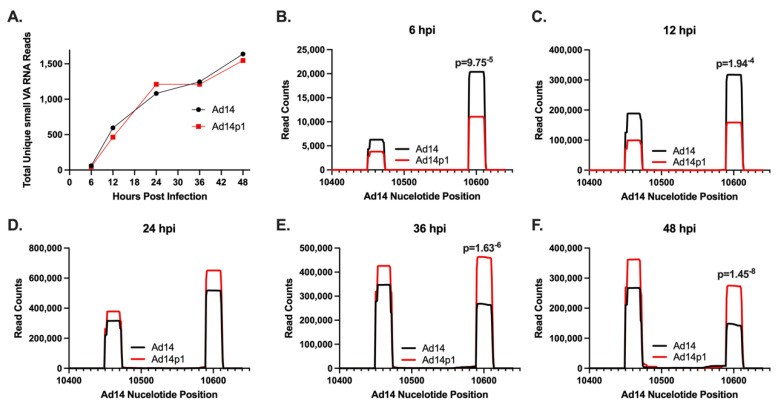
Expression of HAdV14/p1 VA RNA-encoded small RNAs. (**A**) Total number of unique HAdV14 VA RNA-encoded small RNAs during HAdV14 (black) and HAdV14p1 (red) infection of A549 cells. (**B**–**F**) Number of small RNA reads to specific regions of the HAdV14 VA RNA gene during HAdV14 (black) and HAdV14p1 (red) infection. Nucleotides 10,450–10,473 and 10,590–10,613 correspond to the 5′ and 3′ terminal stem regions, respectively. Differential expression was determined with RNA-Seq 2.6 (CLC Genomics Workbench). FDR *p*-values ≤ 0.05 were considered significant and are shown.

**Figure 4 viruses-14-00898-f004:**
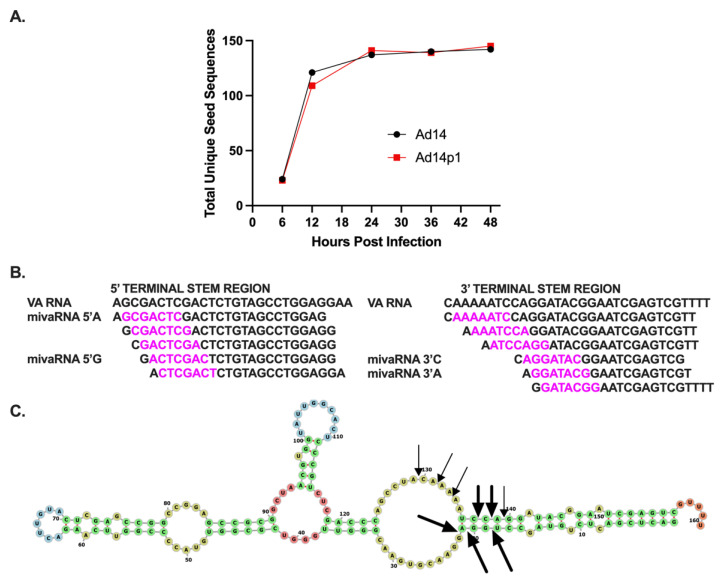
Identification of HAdV14/p1 mivaRNAs. (**A**) Expression of total unique small VA RNA seed sequences during HAdV14 (black) and HAdV14p1 (red) infection of A549 cells. (**B**) Identification of unique seed sequences (purple) from small RNAs that are >1% of total small RNA pool. All aligned to either 5′ or 3′ terminal stem region of VA RNA. The four predominant small RNAs have been designated mivaRNA 5′A, mivaRNA 5′G, mivaRNA 3′C, and mivaRNA 3′A. (**C**) Predicted DICER cleavage sites indicated with bold arrows and regular arrows represent dominant and secondary sites, respectively, in the predicted secondary structure model (RNAStructure).

**Figure 5 viruses-14-00898-f005:**
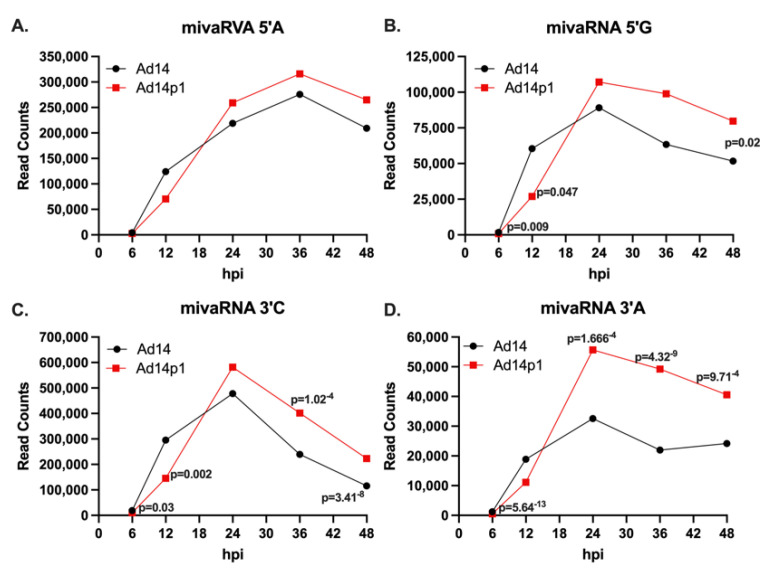
Expression of mivaRNAs during infection. Total read counts of mivaRNA 5′A (**A**), mivaRNA 5′G (**B**), mivaRNA 3′C (**C**), or mivaRNA 3′A (**D**) from 4 replicative infections with either HAdV14 (black) or HAdV14p1 (red) in A549 cells. Differential expression was determined with RNA-Seq 2.6 (CLC Genomics Workbench). FDR *p*-values ≤ 0.05 were considered significant and are shown.

**Figure 6 viruses-14-00898-f006:**
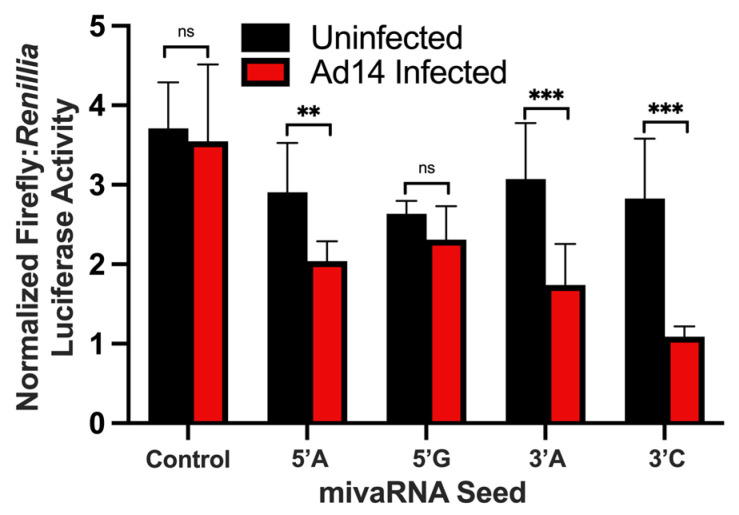
Regulation of luciferase activity by mivaRNAs during viral infection. The complementary sequence of the mivaRNA seeds were cloned into pmirGLO Dual Luciferase Vector and transfected into A549 cells. Twenty-four hours after transfection, A549 cells were infected with Ad14 at an MOI of 10 and luciferase activity was determined 24 h after infection. Control seed is pmirGLO Vector with no mivaRNA complementary sequence. Luciferase activity is expressed as normalized Firefly:Renillia luciferase activity. Mean ± SD, *n* = 8, one-way ANOVA followed by Holm–Sidák test ** *p* < 0.01, *** *p* < 0.001.

**Figure 7 viruses-14-00898-f007:**
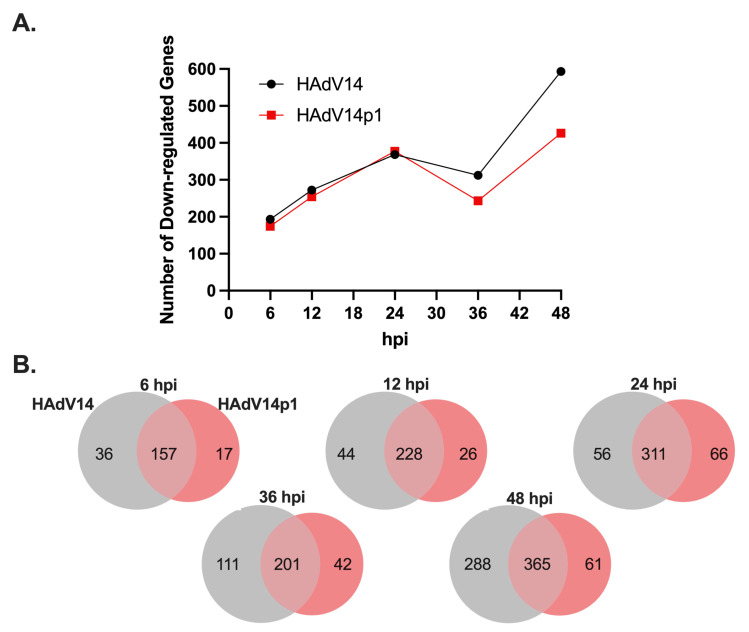
Expression of mivaRNA predicted target genes. (**A**) Number of significantly down regulated (≤−1.5 fold, FDR *p*-value ≤ 0.05) mivaRNA target genes during HAdV14 (black) or HAdV14p1 (red) infection of A549 cells. (**B**) Venn diagram showing shared and unique down regulated target genes during HAdV14 (gray) and HAdV14p1 (red) infection.

**Table 1 viruses-14-00898-t001:** Summary of alignment of sequences reads to the adenovirus 14 genome.

	Mock	6 hpi	12 hpi	24 hpi	36 hpi	48 hpi
**HAdV14**						
Total Reads ^a^	2,915,019	3,408,959	3,391,732	3,387,175	3,330,665	2,821,104
Align to HAdV14 (%) ^b^	331 (0.011%)	26,863 (0.78)	518,023 (15.27)	847,192 (25.01)	633,407 (19.01)	450,635 (15.97)
Align to VA RNA (%) ^c^	300 (90.63%)	26,676 (99.3)	506,467 (97.77)	835,406 (98.61)	617,366 (97.48)	419,037 (92.98)
**HAdV14p1**						
Total Reads		2,803,990	3,267,888	4,990,604	4,160,615	3,310,553
Align to HAdV14 (%) ^b^		14,908 (0.53)	265,128 (8.11)	1,049,717 (21.03)	910,844 (21.89)	700,060 (21.14)
Align to VA RNA (%) ^c^		14,835 (99.51)	258,305 (97.43)	1,032,127 (98.32)	891,497 (97.88)	640,156 (91.44)

^a^ Cumulative from 4 replicate infections and after discarded reads. ^b^ Percentage of total reads that aligned to HAdV14 deWit genome. ^c^ Percentage of HAdv14 aligned reads that mapped to VA RNA gene.

**Table 2 viruses-14-00898-t002:** Expression of HAdV14/p1-encoded small RNAs that are greater than 1% of total small RNA pool.

Virus	Location ^a^	Sequence Reads ^b^						Percentage of Total Pool ^c^					
		Mock	6 hpi	12 hpi	24 hpi	36 hpi	48 hpi	Mock	6 hpi	12 hpi	24 hpi	36 hpi	48 hpi
Ad14	10,450–10,470	58	412	15,526	46,832	107,310	102,582	1.5	3.0	5.6	17.2	23.5	21
	10,450–10,471	4	209	3774	5262	6931	7903	0.8	0.7	0.6	1.1	1.8	22
	10,450–10,472	62	3525	98,392	154,321	148,213	91,456	13.1	19.0	18.3	23.7	20.9	23
	10,450–10,473	3	4	4700	8284	7024	2333	0.0	0.9	1.0	1.1	0.5	24
	10,453–10,472	1	21	252	3790	2889	3324	0.1	0.0	0.4	0.5	0.8	20
	10,453–10,473	35	1698	57,915	81,546	58,245	43,865	6.3	11.2	9.7	9.3	10.0	21
	10,590–10,610	21	1162	38,666	85,568	48,070	20,493	4.3	7.5	10.1	7.7	4.7	21
	10,590–10,611	49	9081	130,289	203,709	106,511	65,227	33.8	25.2	24.2	17.0	14.9	22
	10,590–10,612	21	8172	113,171	167,917	75,507	26,921	30.4	21.8	19.9	12.1	6.2	23
	10,590–10,613	0	622	12,171	17,574	6890	1498	2.3	2.3	2.1	1.1	0.3	24
	10,591–10,612	5	323	4579	7136	6369	9296	1.2	0.9	0.8	1.0	2.1	22
	10,591–10,613	11	847	13,522	23,972	14,593	13,366	3.2	2.6	2.8	2.3	3.1	23
Ad14p1	10,450–10,470	58	312	9946	53,273	97,088	100,549	2.1	3.8	5.1	10.8	14.6	21
	10,450–10,471	4	147	2433	6261	7252	11,277	1.0	0.9	0.6	0.8	1.6	22
	10,450–10,472	62	2257	54,642	180,877	193,469	144,571	15.2	20.7	17.3	21.4	21.1	23
	10,450–10,473	3	41	2421	12,498	10,975	3347	0.3	0.9	1.2	1.2	0.5	24
	10,453–10,472	1	16	598	4144	6188	7838	0.1	0.2	0.4	0.7	1.1	20
	10,453–10,473	35	823	26,019	99,068	90,307	66,737	5.5	9.9	9.5	10.0	9.7	21
	10,590–10,610	21	630	15,935	94,003	82,196	46,081	4.2	6.0	9.0	9.1	6.7	21
	10,590–10,611	49	5107	65,870	238,825	167,640	122,769	34.3	24.9	22.9	18.6	17.9	22
	10,590–10,612	21	4369	57,370	222,279	135,923	45,857	29.3	21.7	21.3	15.1	6.7	23
	10,590–10,613	0	348	6097	23,323	12,092	2321	2.3	2.3	2.2	1.3	0.3	24
	10,591–10,612	5	133	2845	11,361	11,868	13,477	0.9	1.1	1.1	1.3	2.0	22
	10,591–10,613	11	368	7849	42,258	35,445	24,720	2.5	3.0	4.1	3.9	3.6	23

^a^ Numbers refer to location in HAdV14 genome accession number. ^b^ Cumulative reads from 4 independent infections. ^c^ Percentage of reads from the total small RNA reads.

**Table 3 viruses-14-00898-t003:** Unique miRNA seed sequences encoded in the HAdV14/p1 VA RNA.

Seed ^a^	Count ^b^	Seed	Count	Seed	Count	Seed	Count
AGGATAC	1,148,067	CCCAGCC	253	CGGTGTA	90	GGGTTGG	34
GCGACTC	835,530	CCAGCCT	252	GGTGTAC	90	TAGCCTG	33
ACTCGAC	272,742	GCCTACA	240	TGTACTC	90	AGCCTGG	31
GGATACG	98,828	TTGTACT	237	TGGTATT	86	CGCGGCT	31
GACTCGA	9983	GACTTGT	232	TTGGCAC	83	GAGCCGG	31
GATACGG	5133	TGGCACT	232	GGGTCGC	72	GAGCCGC	28
CTCGACT	3461	TCGACTC	228	CTGTAGC	71	GGTTGGG	28
CGACTCG	2120	GCGGTGT	220	CTTGTAC	71	AACGTGG	27
AAAAATC	1671	CCCCGGT	217	TGTACCC	69	TTGGGTC	26
ACTCCCG	1557	CCCGGTT	174	CTCGAGC	68	ACGTGGT	25
AAATCCA	989	CGGCCGG	173	ACTCGAG	67	GTATTGG	20
ATACGGA	923	CGTCTCG	173	GTACCCC	62	CCGGAGC	19
CCCGTCT	821	TTCAAGA	168	GGAGCCG	61	AGCCGCG	18
CCGTCTC	814	TGTAGCC	165	ACGGGTT	59	TGGAGGA	18
ATCCAGG	789	TCTCGAC	161	TCGAGCC	59	TCGCGGT	17
AGCCTAC	766	CAGCCTA	156	GACTCTG	58	CGGGTTG	15
AATCCAG	547	GGCACTC	155	GCCGGAG	58	GAACGGG	15
TCCAGGA	479	ACCCCGG	148	GGTATTG	58	GCCGCGG	15
CTCCCGT	418	CCGGTTC	146	ACTTGTA	57	AGCCGGC	14
TCAAGAC	379	CTACAAA	145	CTCTGTA	56	CGGCTAA	14
TACAAAA	371	CGACTCT	143	GTCGCGG	55	GCCGGCC	14
CGACCCA	362	GGCCGGA	136	GTTCAAG	52	CTAACGT	12
AGACTTG	355	GTAGCCT	132	CGAGCCG	51	GCCTGGA	12
GTGGTAT	352	GGTCGCG	129	TAACGTG	51	GTGAACG	12
CAAAAAT	339	GTGTACC	126	TCTGTAG	48	GGAACGT	10
ACCCAGC	331	GCACTCC	122	CGCGGTG	47	CCTGGAG	8
ACAAAAA	323	TACTCGA	120	TATTGGC	46	CTGGAGG	8
AAAATCC	319	CCAGGAT	119	ACTCTGT	44	TGAACGG	7
GACCCAG	318	ATTGGCA	115	GCGGCTA	39	ACGTGAA	6
CAAGACT	317	CGGAGCC	111	GGCTAAC	39	CGTGAAC	5
GTCTCGA	310	TACCCCG	105	CGTGGTA	38	GCTAACG	5
CAGGATA	304	TACGGAA	104	GTTGGGT	37	AACGGGT	4
CACTCCC	300	CGGTTCA	95	GTACTCG	35	AGGAACG	4
AAGACTT	289	CCGGCCG	94	TGGGTCG	35	GAGGAAC	3
CCTACAA	288	GGTTCAA	94	GGAGGAA	34	AACGTGA	2

^a^ Nucleotides 2–8 of small RNAs that mapped to the HAdV14 VA RNA gene. ^b^ Total read counts from 4 replicative infections and all time points.

**Table 4 viruses-14-00898-t004:** Summary of sequence reads that map to miRbase or HAdV14 VA RNA.

	Mock	6 hpi	12 hpi	24 hpi	36 hpi	48 hpi
HAdV14						
Total Reads ^a^	2,915,019	3,408,959	3,391,732	3,387,175	3,330,665	2,821,104
miRBase (%) ^b^	1,794,862 (61.57%)	2,198,516 (64.49)	1,628,255 (48.01)	1,389,443 (41.02)	1,277,346 (38.35)	1,376,789 (48.8)
Align to VA RNA (%) ^c^	300 (0.013)	26,676 (0.78)	506,467 (14.93)	835,406 (24.63)	617,366 (18.54)	419,037 (14.85)
HAdV14p1						
Total Reads		2,803,990	3,267,888	4,990,604	4,160,615	3,310,553
miRBase (%) ^b^		1,755,693 (62.61)	1,533,510 (46.93)	1,738,608 (34.84)	1,507,889 (36.24)	1,242,118 (37.52)
Align to VA RNA (%) ^c^		14,835 (0.53)	258,305 (7.90)	1,032,127 (20.68)	891,497 (21.42)	640,156 (19.34)

^a^ Cumulative from 4 replicate infections and after discarded reads. ^b^ Percentage of total reads that mapped to miRbase V22. ^c^ Percentage of total reads that aligned to VA RNA.

**Table 5 viruses-14-00898-t005:** Cellular miRNA and mivaRNA expression during HAdV14 infection.

Uninfected		HAdV14 6 hpi		HAdV14 12 hpi		HAdV14 24 hpi		HAdV14 36 hpi		HAdV14 48 hpi	
miRNA	Counts ^a^	miRNA	Counts	miRNA	Counts	miRNA	Counts	miRNA	Counts	miRNA	Counts
hsa-miR-21-5p	378,165	hsa-miR-21-5p	501,953	hsa-miR-21-5p	341,239	mivaRNA 3’C	478,333	mivaRNA 5’A	275,804	hsa-miR-21-5p	234,917
hsa-miR-192-5p	124,241	hsa-miR-192-5p	226,084	mivaRNA 3’C	295,421	hsa-miR-21-5p	291,813	mivaRNA 3’C	239,471	mivaRNA 5’A	209,207
hsa-miR-22-3p	110,256	hsa-miR-22-3p	112,411	hsa-miR-192-5p	164,717	mivaRNA 5’A	218,763	hsa-miR-21-5p	238,252	mivaRNA 3’C	115,779
hsa-let-7a-5p	57,562	hsa-miR-30a-5p	72,082	mivaRNA 5’A	124,246	hsa-miR-192-5p	140,484	hsa-miR-192-5p	142,357	hsa-miR-192-5p	104,677
hsa-miR-30a-5p	56,430	hsa-let-7a-5p	60,804	hsa-miR-22-3p	83,424	mivaRNA 5’G	89,099	mivaRNA 5’G	63,339	hsa-miR-22-3p	73,414
hsa-miR-27b-3p	53,325	hsa-miR-181a-5p	54,302	mivaRNA 5’G	60,491	hsa-miR-22-3p	69,931	hsa-miR-22-3p	51,216	mivaRNA 5’G	51,755
hsa-miR-21-3p	35,247	hsa-miR-182-5p	44,720	hsa-miR-30a-5p	48,290	hsa-miR-30a-5p	40,481	hsa-miR-181a-5p	42,803	hsa-let-7a-5p	41,100
hsa-miR-182-5p	31,774	hsa-miR-27b-3p	40,379	hsa-let-7a-5p	45,887	hsa-miR-181a-5p	37,262	hsa-let-7a-5p	32,618	hsa-miR-181a-5p	34,290
hsa-miR-181a-5p	30,030	hsa-miR-92a-3p	39,737	hsa-miR-92a-3p	40,458	hsa-let-7a-5p	35,427	hsa-miR-30a-5p	32,348	hsa-miR-92a-3p	30,307
hsa-let-7f-5p	21,049	hsa-miR-21-3p	36,995	hsa-miR-181a-5p	39,273	mivaRNA 3’A	32,571	hsa-miR-182-5p	29,411	mivaRNA 3’A	24,185
hsa-miR-92a-3p	20,949	hsa-miR-26a-5p	25,407	hsa-miR-182-5p	37,039	hsa-miR-92a-3p	32,411	hsa-miR-92a-3p	27,288	hsa-miR-27b-3p	23,164
hsa-miR-26a-5p	20,753	hsa-miR-31-5p	24,588	hsa-miR-21-3p	28,190	hsa-miR-182-5p	28,844	hsa-miR-21-3p	25,343	hsa-let-7f-5p	21,635
hsa-miR-30d-5p	17,685	hsa-let-7f-5p	22,516	hsa-miR-27b-3p	27,473	hsa-miR-21-3p	22,806	mivaRNA 3’A	21,998	hsa-miR-182-5p	20,777
hsa-miR-191-5p	16,446	hsa-miR-30d-5p	20,889	mivaRNA 3’A	18,853	hsa-miR-27b-3p	19,105	hsa-miR-27b-3p	17,069	hsa-miR-30a-5p	20,767
hsa-miR-31-5p	16,293	hsa-miR-191-5p	19,910	hsa-miR-31-5p	18,286	hsa-let-7f-5p	16,025	hsa-let-7f-5p	14,244	hsa-miR-21-3p	20,269
hsa-miR-10a-5p	16,041	mivaRNA 3’C	19,063	hsa-let-7f-5p	18,159	hsa-miR-26a-5p	13,932	hsa-miR-191-5p	11,325	hsa-miR-26a-5p	14,845
hsa-let-7i-5p	12,174	hsa-miR-10a-5p	15,222	hsa-miR-30d-5p	17,511	hsa-miR-31-5p	12,925	hsa-miR-26a-5p	10,941	hsa-miR-191-5p	10,567
hsa-miR-16-5p	10,501	hsa-miR-151a-3p	13,537	hsa-miR-26a-5p	17,150	hsa-miR-30d-5p	12,231	hsa-miR-30d-5p	9232	hsa-miR-31-5p	10,283
hsa-let-7e-5p	10,261	hsa-miR-28-3p	12,869	hsa-miR-191-5p	16,866	hsa-miR-191-5p	11,989	hsa-miR-31-5p	9162	hsa-miR-16-5p	9116
hsa-miR-125a-5p	10,223	hsa-let-7i-5p	10,668	hsa-miR-151a-3p	12,185	hsa-miR-151a-3p	9331	hsa-miR-151a-3p	8393	hsa-miR-10a-5p	8900

^a^ Cumulative from 4 replicate infections.

**Table 6 viruses-14-00898-t006:** Enrichment of mivaRNA targets by KEGG analysis.

Cellular Pathway ^a^ (# of Targets) ^b^	
Metabolic Pathways (141)	Axon Guidance (33)
Pathways in Cancer (73)	Breast Cancer (32)
PI3K-Akt Signaling (49)	Rap1 Signaling (32)
Pathways in Neurodegeneration (42)	Focal Adhesion (32)
Proteoglycans in Cancer (41)	mTOR Signaling (31)
Herpes Simplex Virus 1 Infection (40)	Calcium Signaling (31)
Human Papillomavirus Infection (40)	Ras Signaling (31)
MAPK Signaling (37)	Wnt Signaling (29)
Regulation of Actin Cytoskeleton (36)	Shigellosis (29)
cAMP Signaling (35)	Oxytocin Signaling (29)

^a^ Pathway, diseases, functions in Homo sapiens. ^b^ Number of mivaRNA predicted target genes involved in pathways.

**Table 7 viruses-14-00898-t007:** IPA enrichment of top 10 canonical pathways at 36 hpi ^a^.

HAdV14	HAdV14p1
Glioma Signaling	Glioma Signaling
Molecular Mechanisms of Cancer	Molecular Mechanisms of Cancer
Senescence Pathway	Hepatic Fibrosis Signaling Pathway
Apelin Endothelial Signaling Pathway	CXCR4 Signaling
Epithelial Adherens Junction Signaling	Glioblastoma Multiforme Signaling
CXCR4 Signaling	IGF-1 Signaling
NF-κB Activation by Viruses	Macropinocytosis Signaling
Glioblastoma Multiforme Signaling	Epithelial Adherens Junction Signaling
Hepatic Fibrosis Signaling Pathway	NF-κB Activation by Viruses
Estrogen Receptor Signaling	Osteoarthritis Pathway

^a^ Ranked by highest −log (*p*-value) assigned by IPA.

## Data Availability

All RNA-seq data have been deposited to the Sequence Read Archive (SRA). Both the small RNA-seq and RNA-seq data are available from the bioproject PRJNA752359.

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
