# Peer review of "Characterization of Viral miRNAs during Adenovirus 14 Infection and Their Differential Expression in the Emergent Strain Adenovirus 14p1"

_viruses, 2022, doi:10.3390/v14050898_

Round 1
Reviewer 1 Report
In this manuscript, McIndoo et al. examine the expression of miRNAs following Adenovirus 14 infection from the virus-associated RNAs (VA RNAI). The authors find differential expression patterns of 5’ and 3’ miRNAs from the VA RNAI that may impact host gene expression during infection. Overall, the manuscript is well written and increases our knowledge of the produced viral miRNAs that might affect infection.
Major Concerns:
- The authors examine differences in the expression of VA RNA encoded small RNAs in figure 3. While the authors find statistical differences in the expression between HAdV-14 and the newly emerging p1 strain, it is unclear if these two viruses infect A549 cells with similar kinetics. The authors should provide a growth curve or reference previous studies so that we can adequately judge the differences in small RNA expression. For example, are these fundamental differences, or are viral miRNA expression patterns different due to changes in viral attachment, entry, transcription, etc., between the two strains?
- The authors use cumulative data (or reads) from four replicate infections. It may be better to show the average number of counts from these reads. If the number of reads for a specific miRNA were over-represented in one of the experiments, it would skew the interpretation of the data. The authors should at least address why cumulative data were used.
- The authors conclude that the identified viral miRNAs impact several host genes. However, their experiments were in the context of a full HAdV infection with viral proteins that have been shown to influence the cellular pathways identified in their screen. It is unclear if these viral miRNAs directly impact genes in these pathways without further experimentation.
Minor Concerns:
- A minor concern, but a few of the tables have letters that need to be moved to a superscript—for example, the labels in Table 2.
- Minor spelling error on line 249: “Seequences.”
- The authors mention in lines 334 and 335 that they identified a viral miRNA that had the same seed sequence as a host miRNA. Is the target for this miRNA characterized?
- Line 262-264, the authors state, “ we predict that there are 4 dominant Dicer cleavage sites…” It is unclear how the authors predicted these sites. The authors may want to elaborate just a little more for reader clarity.
Author Response
Reviewer #1. We would like to thank this reviewer for their critical assessment of our manuscript and their constructive remarks which will improve the work.
Major Concern #1: We have addressed the potential for differences in viral replication creating the differential expression of miRNA in the discussion in lines 541-550. In brief, both Anderson et al. and our lab have shown that there is no difference between HAdV14 and HAdV14p1 replication in either tissue culture cells or in the lungs of Syrian hamsters. Second, which isn’t mentioned in the text, we have previously observed that rates of CPE are identical between both viruses at the exact MOI used for these studies. These results indicate that viral attachment, entry or replication rates do not explain the observed differences in mivaRNA expression.
Major Concern #2: Cumulative data were used for two reasons. First, the cumulative data is appropriate to use for identifying the major mivaRNAs produced during viral infection as well as the key mivaRNA seed sequences. Second, in order to avoid excess use of tables we used cumulative data for the graphs in figures 3 and 5. The statistical analysis was performed on normalized data as explained in detail in the methods section. Therefore, we feel the use of cumulative data in these figures is appropriate as all assessment of differential expression is done with normalized data according to accepted standards for assessing differential miRNA expression.
Major Concern #3: “The authors conclude that the identified viral miRNAs impact several host genes.” We never made such conclusions for our results. We did conclude that mivaRNA are viable miRNAs that can regulate gene expression through the use of the luciferase reporter vector. RNA-seq was used to determine whether any of the potential target genes were downregulated during infection. We did observe that some were down-regulated but we never concluded that this was only a result of mivaRNA expression. We agree that our studies do not show that mivaRNAs are the only mechanism through which target gene expression may be regulated. To clarify this point, we have added a sentence in line 492 stating that. We are planning such experiments, but those studies are out of the scope and focus of this paper.
Minor Concerns
1.) We have corrected the superscript issue. That appeared to be a formatting resulting from moving the tables from one word document to the journal’s template.
2.) Spell check was performed to correct a few errors.
3.) Targets for miR-584c-3p have been predicted.
4.) We agree this needed clarification. We have expanded on this statement, starting on line 318.
Reviewer 2 Report
Manuscript by McIndoo et al., deals with HAdV14 non-coding RNA named as VA RNA and more particularly with VA RNA-derived miRNAs (mivaRNAs). VA RNAs and mivaRNAs are well characterized for the prototypic HAdV2/5, however, basically nothing is known about the HAdV14 encoded VA RNA and mivaRNAs. In this regards the manuscript provides novel and interesting data. Similarly to other HAdVs, the authors found 4 dominant mivaRNAs generated from the HAdV14/p1 VA RNA gene, two from each of the 5’ and 3’ regions of the terminal stem. Further they show unique temporal expression pattern of mivaRNAs as well as that the encoded mivaRNAs are functional, although their exact targets on cellular mRNAs were not validated.
The manuscript is well written, methods are described in details, experiments are well-controlled. I would lift up the discussion part as the authors nicely discuss their work in connection to the previous mivaRNA studies.
I am very confident that this is the tip of the iceberg published from the authors study, taken into consideration that RNA-Seq generates enormous amount of data. So, the present manuscript is an elegant introduction into potential future manuscripts how pathogenic HAdV14 infection alters small RNA metabolism in the cells.
In addition there are following minor comments:
L219: mivaRNA 5'C has to be mivaRNA 5'G
Fig. 4C: the program used to remodel HAdV14 VA RNA should be mentioned in the figure legend. Would be nice to have a prototypical HAdV2/5 VA RNA I on the side to compare the structures of these two. However, it is not 100% must, although it will be informative.
Fig 6: what is the control? pmirGLO without complementary mivaRNA sequence? This information has to be written out in the figure legend.
L370: RNA Pol has to defined as RNA Pol III
Discussion: HAdV37 infection generates a unique viral small RNA called MLP-TSS-sRNA (Kamel & Akusjärvi, 2017). Did the authors find similar sRNA in HAdV14? This should be mentioned in the discussion. Also the fact that mivaRNAs have been found in HAdV+ patient tonsillar T lymphocytes (Assadian., 2017) should be mentioned (for example in connection to L440).
Check also spelling: Dicer vs. DICER
Author Response
Reviewer #2. We would like to thank this reviewer for their critical assessment of our manuscript and their constructive remarks which will improve the work. As noted, this is just the tip of the iceberg. We are preparing a manuscript exploring the effects of Ad14 and Ad14p1 on host miRNA expression, and the teaser is that there is some differential expression of host miRNA between the two viruses!! These studies provide the basis for our ongoing analysis of VA RNA activities in Ad pathogenesis.
Minor Concerns:
1.) We have fixed the typo of mivaRNA 5’C to 5’G as suggested.
2.) We have added the source of the model to figure 4. It was mentioned previously in the figure 1 legend. While we understand the interest in adding a model of HAdV2/5 VA RNA to the figure for comparisons, we have no experimental data to support the predicted secondary structure for HAdV14 VA RNA and, therefore, do not think it would be appropriate to try to compare the models, side by side. Such comparisons would require future data on stem-loop structures.
3.) Fig6 Legend. We have added that the control is without complementary mivaRNA sequence to the legend.
4.) RNA Pol has been corrected to RNA Pol III as suggested.
5.) We have not looked yet for the presence of a MLP-TSS-sRNA in HAdV14. That is on our list to do. We do know that we have small RNAs from all over the HAdV14 genome. Nearly all represent just small numbers of reads and are dwarfed by the VA RNA reads. But we are searching for any other leads. We are also working on improving the HAdV14 annotation (through direct long read RNA sequencing) as it is incomplete and makes bioinformatic analysis difficult. We have added a sentence (line 550) talking about the MLP-TSS-sRNA as well as including the reference on mivaRNAs found in latent infected tonsillar lymphocytes.
Round 2
Reviewer 2 Report
The authors have answered to all of my questions and have implemented corrections int the manuscript. Very well done and thank you for the teaser!